# Blocking IL-17: A Promising Strategy in the Treatment of Systemic Rheumatic Diseases

**DOI:** 10.3390/ijms21197100

**Published:** 2020-09-26

**Authors:** Carlos Rafael-Vidal, Nair Pérez, Irene Altabás, Samuel Garcia, Jose M. Pego-Reigosa

**Affiliations:** 1Rheumatology & Immuno-Mediated Diseases Research Group (IRIDIS), Galicia Sur Health Research Institute (IIS Galicia Sur). SERGAS-UVIGO, 36201 Vigo, Spain; carlos.rafael@iisgaliciasur.es (C.R.-V.); nair.pg.89@gmail.com (N.P.); irene.altabas@gmail.com (I.A.); jose.maria.pego.reigosa@sergas.es (J.M.P.-R.); 2Rheumatology Department, University Hospital Complex of Vigo, 36201 Vigo, Spain

**Keywords:** Interleukin-17, systemic rheumatic diseases, T helper 17 cells, Sjögren’s syndrome, systemic lupus erythematosus, systemic sclerosis, therapeutic target

## Abstract

Systemic rheumatic diseases are a heterogeneous group of autoimmune disorders that affect the connective tissue, characterized by the involvement of multiple organs, leading to disability, organ failure and premature mortality. Despite the advances in recent years, the therapeutic options for these diseases are still limited and some patients do not respond to the current treatments. Interleukin-17 (IL-17) is a cytokine essential in the defense against extracellular bacteria and fungi. Disruption of IL-17 homeostasis has been associated with the development and progression of rheumatic diseases, and the approval of different biological therapies targeting IL-17 for the treatment of psoriatic arthritis (PsA) and ankylosing spondylitis (AS) has highlighted the key role of this cytokine. IL-17 has been also implicated in the pathogenesis of systemic rheumatic diseases, including systemic lupus erythematosus (SLE), Sjögren’s syndrome (SS) and systemic sclerosis (SSc). The aim of this review is to summarize and discuss the most recent findings about the pathogenic role of IL-17 in systemic rheumatic and its potential use as a therapeutic option.

## 1. Introduction

Systemic rheumatic diseases are chronic, inflammatory, autoimmune disorders that may affect every organ system. Similar to other autoimmune pathologies, systemic rheumatic diseases are characterized by the loss of homeostatic tolerance of the immune system, which is a consequence of the abnormal activation against self-antigens and leads to the production of autoantibodies, inflammatory mediators and immune complexes. All these processes trigger local and systemic inflammation that induce damage to the affected organs and finally lead to disability, loss of quality of life and premature death [1,2,3]. The best-characterized systemic rheumatic diseases are likely rheumatoid arthritis (RA), systemic lupus erythematosus (SLE), or Sjögren’s syndrome (SS), but there are also other less frequent disorders, such as systemic sclerosis (SSc), vasculitis or immunoglobulin G4–related disease (IgG4-RD).

Cytokines are soluble mediators that have an essential role in immune functions, and the dysregulation of homeostatic cytokine levels has been associated with the pathogenesis of autoimmune diseases [4]. The interleukin-17 (IL-17) family is a group of pro-inflammatory cytokines with a key role in defense against fungi and bacteria. Elevated IL-17 levels have been associated with the development of chronic inflammatory immune-mediated diseases [5,6,7] and several therapeutic strategies targeting IL-17 are approved for the treatment of psoriasis, psoriatic arthritis (PsA) and ankylosing spondylitis (AS) [8], pointing to the role of this cytokine in the pathogenesis of these autoimmune diseases [9,10,11]. The role of IL-17 on RA has been extensively studied [7,12,13]; therefore, this review will focus on the role of IL-17 in the pathogenesis of other systemic rheumatic diseases, mainly SLE, SS and SSc, as well as the potential therapeutic intervention on these diseases.

## 2. IL-17 Expression and Function

The IL-17 family consists of six family members: IL-17A (also named IL-17), which was the first member reported [14], IL17B, IL-17C, IL-17D, IL-17E and IL-17F [15]. IL-17A has a predominant role in the pathogenesis of autoimmune diseases, and therefore in this review we will focus on this member [6].

Several cell types (cytotoxic T 17 cells (Tc17), γδ T cells, invariant natural killer T cells (iNKT), group 3 innate lymphoid cells (ILC3), natural killer (NK) cells and double-negative T cells [16,17]) are able to produce IL-17, although T helper 17 (Th17) cells are the best-characterized IL-17-producing cell type [18,19]. Th17 cells are a subset of CD4^+^ T helper cells that are characterized by the production of the cytokines IL-17 and IL-22, but also secrete other important cytokines such as IL-21 and tumor necrosis factor (TNF). Th17 cells are essential for the host defense against fungi and bacteria, and the excessive activation of Th17 cells is associated with the pathology of multiple sclerosis (MS) [20,21], inflammatory bowel disease (IBD) [22,23], systemic sclerosis (SSc) [24,25], psoriasis [26,27], rheumatoid arthritis (RA) [28] and spondyloarthropathies [29,30].

Multiple works have shown that IL-23 is a key cytokine involved in Th17 differentiation and in the production of IL-17 by other cell types (iNKT, NK, γδ T cells, CD8^+^ T cells) [31,32]. IL-23, through binding to its receptor IL-23R, promotes the phosphorylation of the signal transducer and activator of transcription 3 (STAT3) by janus kinase 2 (JAK2) and tyrosine kinase 2 (TYK2), allowing its entrance into the nucleus and enhancing the expression of the retinoic acid receptor-related orphan receptor gamma t (RORγt), which is responsible for the expression of IL-17 and other Th17 cytokines [12,33]. Although IL-23 is the main cytokine involved in Th17 differentiation, other inflammatory mediators, such as IL-21 [34,35], IL-1 [36], TGF-β [37] and IL-6 [38], are essential for proper cell differentiation.

IL-17 signaling occurs through specific membrane receptors: the IL-17 family receptors (IL-17R). There are five IL-17R members, which are heterodimers consisting of two different subunits: IL-17RA/IL-17RC, the receptor for IL-17A [39,40,41]. The binding of IL-17 to its receptor activates two different pathways, the canonical and the non-canonical. In the canonical pathway, the binding of IL-17 to the IL17RA/IL17RC complex induces the binding of the adaptor protein Act1 to the SEFIR (similar expression of fibroblast growth factor genes/IL-17 Receptor) complex, which triggers the ubiquitylation of TNF-receptor-associated factor-6 (TRAF6) and the activation of mitogen-activated protein kinase (MAPK), CCAAT-Enhancer-Binding protein β/δ (C/EBPβ/δ), and nuclear factor κB (NF-κB) pathways [15,42]. These pathways promote the production of antimicrobial peptides and proinflammatory cytokines and chemokines [15,42,43].

On the other hand, the non-canonical pathway promotes the stabilization of m(essenger)RNAs encoding for inflammatory mediators. In this pathway, IL-17 induces the phosphorylation of Act1 that leads to the inhibition of splicing factor 2 (SF2), which is a factor involved in mRNA destabilization, and to the recruitment of the mRNA stabilizer human antigen R (HuR), which induces the stabilization of the mRNA and the secretion of inflammatory cytokines and chemokines [15,43,44,45].

Regarding the biological function, IL-17 is a cytokine with a crucial role in the host defense against the pathogen infections, mainly against extracellular bacteria and fungi on epithelial and mucosal surfaces [31]. IL-17 acts on fibroblasts, immune cells and epithelial cells, inducing the production of antimicrobial molecules, cytokines, chemokines and matrix metalloproteinases (MMPs) [15,17,42]. The secretion of these mediators promotes the recruitment of neutrophils and other immune cells, like Th17 themselves and ILC3s, to the affected tissues, producing an effective and protective immune response. In addition, IL-17 is also involved in the activation of B cells and the formation of germinal centers [15,43,46,47,48].

## 3. IL-17 in Systemic Rheumatic Diseases

### 3.1. Systemic Lupus Erythematosus

Systemic lupus erythematosus (SLE) is a heterogeneous autoimmune disease that predominantly affects women of reproductive age and is characterized by the production of autoantibodies and the formation and deposition of immune complexes that induce systemic inflammation and tissue damage. SLE can affect almost any organ, although the most commonly affected are skin and joints. SLE is a complex multifactorial disease and environmental, genetic and immunologic factors are involved in its development [49,50]. Although some pathogenic mechanisms of SLE are still unknown, it has been widely reported that the main immunological events involved in the pathogenesis of SLE are the disbalance between apoptotic material production and its removal, the type I interferon (IFN) altered signaling and the T and B cells’ abnormal activation, which lead to autoantibody production.

Different human and mice studies have demonstrated a dysregulation of IL-17 in SLE and its role in this disease. The levels of IL-17 and the frequency of IL-17-producing cells were elevated in two different lupus mouse models [51] and, importantly, the genetic deletion of IL-17 was shown to ameliorate the pathology of SLE [52]. Regarding IL-17 studies in humans, the levels of IL-17 producing cells were increased in the peripheral blood of SLE patients [53,54,55] as well as in target tissues like kidneys [56]. Circulating levels of IL-17A are also elevated in patients with SLE compared to HC [57,58,59]. The association between these levels and the disease activity is controversial, as Chen et al., Wong et al. and Abdel Galil et al. found a positive correlation between IL-17 levels and the activity of the disease measured by the SLE Disease Activity Index (SLEDAI), but these correlations were not observed in a further study [60]. Therefore, the use of IL-17 as a biomarker of disease activity needs to be elucidated in further studies. IL-17A levels were also positively correlated with the RORγt mRNA expression in SLE patients, pointing out the relevance of the Th17-IL-17 axis in this disease [57]. This assumption is supported by the fact that IL-23 and IL-21, two cytokines involved in Th17 differentiation, are also elevated in SLE patients [58,61,62].

There is important evidence showing that IL-17 has a pleiotropic role on SLE pathology [7]. IL-17 induces the recruitment of neutrophils and other immune cells to the target tissues, promoting and maintaining the inflammatory process. The IL-17-induced neutrophil recruitment is an important effect, as neutrophils are key cells in SLE pathology. The low-density granulocytes, a subpopulation of neutrophils that are prone to cell death, have been seen to contribute to the pathology of this disease, through the Neutrophil Extracellular Traps formation (NETosis) process, in which the release of intracellular material to the surrounding milieu contributes to the initiation and maintenance of immunological alteration in SLE [63,64]. In fact, the cellular debris released by these cells induce the activation of the type I IFN pathway by plasmacytoid dendritic cells (pDCs), which ultimately leads to the hyperactivation of T and B cells [49,54], consequently perpetuating the characteristic inflammatory processes of SLE. Remarkably, it has been demonstrated that IL-17 is able to induce NETosis in SLE in an animal model of lupus [65].

IL-17 also induces the production of pro-inflammatory cytokines by different cell types. An essential IL-17-induced cytokine is IL-6, which is produced by macrophages and monocytes, and also in SLE patients by B cells. Among the roles of IL-6 on SLE pathogenesis, its effect on B cells’ differentiation into plasma cells and the consequent production of autoantibodies is the most relevant [59]. Furthermore, as IL-6 promotes the differentiation of Th17 cells, it has been shown that the IL-17/IL-6 axis induces a positive feedback loop in SLE [66]. Finally, IL-17 also induces the survival, proliferation and differentiation of B cells, generating the subsequent production of autoantibodies [67].

Altogether, these works demonstrate that IL-17 is involved, in several ways, in the development and onset of SLE, and therefore targeting IL-17 might be a promising therapeutic option.

### 3.2. Systemic Sclerosis

Systemic sclerosis (SSc), also known as scleroderma, is a chronic autoimmune disease that may involve virtually any organ system, and whose mortality rate is the highest among the rheumatic diseases. Similarly to other systemic autoimmune diseases, genetic, environmental and immunological factors are involved in its pathogenesis. SSc is characterized by the inflammation, dysregulation of innate and adaptive immune systems, vascular abnormalities, and fibrosis, which is the hallmark of this disease and is due to an excessive production of collagen in several organs, mainly skin and lungs [68,69].

Several studies have shown a dysregulation of IL-17 homeostasis in SSc patients, but there are certain discrepancies about the levels at which IL-17 is found in patients with SSc. Despite some authors indicating that IL-17A is not increased in SSc patients [24,70], other studies have found that circulating IL-17 levels are elevated in patients with SSc [25,71]. These differences may be attributed to the difficulty of measuring IL-17 systemically, as the circulating levels are really low. However, the elevated frequency of Th17 cells and IL-17^+^ cells found in the peripheral blood and skin in SSc patients [25], as well their association with the skin thickening, demonstrate the deregulated IL-17 signaling and the potential implication in SSc pathology [24,25,72,73,74]. On the other hand, and analogously to what occurs in SLE, increased levels of IL-23 and other cytokines that promote Th17 differentiation (IL-1, IL-6, TGF-β and IL-21) have been found in SSc patients [75,76].

The works about the role of IL-17 on fibrosis showed opposite effects in mice and human studies. In mouse models, it has been widely reported that IL-17 has a pro-fibrotic role. Okamoto and colleages found that IL-17 deficiency reduced skin fibrosis in two SSc mice models: the bleomycin-induced and the TSK-1 model. Moreover, IL-17 stimulation induced a pro-fibrotic phenotype in a fibroblast murine cell line [77,78]. These findings were corroborated in another work in which IL-17 promoted the expression of type I collagen, TGF-β and IL-6 by pulmonary mouse fibroblasts [79]. Moreover, Park et al. showed that IL-1-mediated skin and lung fibrosis was dependent on IL-17 activity [77], pointing out the fibrotic role of mouse IL-17.

Despite the work of Yang et al., which showed that, in dermal fibroblasts of SSc patients, the blockage of IL-17 diminished the expression of Col(lagen) I and III induced by the supernatants of SSc peripheral blood mononuclear cells (PBMCs) and Th17 cells [25]; the profibrotic role of IL-17 was not observed in isolated human fibroblasts. In fact, Brembilla et al. [80] found that IL-17 stimulation did not regulate the expression of Col I by dermal fibroblasts of SSc patients and Carvalheiro et al. [81] showed that IL-17 neutralization did not affect the expression of collagen family members by dermal fibroblasts, by reducing the expression of fibrotic mediators such as IL-6. Remarkably, complex human models appear to have contributed to solving the discrepancies regarding the role of IL-17, as, in an organotypic full human skin model, IL-17 had a strong anti-fibrotic role through the reduction in Col I production and the promotion of the MMP-1 secretion [82].

The role of IL-17 in inflammatory processes observed in SSc patients is well defined. IL-17 induces the production of cytokines (IL-6, IL-1), chemokines (IL-8, CCL-2, CCL-8, CCL-20, CXCL-2), matrix metalloproteinases (MMP-1, MMP-2, MMP-9) and other adhesion molecules by the dermal fibroblast and endothelial cells (ECs) of SSc patients, which perpetuate the inflammatory processes and, as mentioned above, contribute to fibrosis and vascular dysfunction-related processes [71,75,76,81,82].

Another essential aspect of SSc in which IL-17 has been implicated is vascular thickening and endothelial dysfunction. Besides the induction of inflammatory mediators by ECs, it has been reported that IL-17 activates and regulates the proliferation and migration of vascular smooth muscle cells, which are key in vascular pathology in SSc since, among other functions, they promote the production of collagen [76,83].

In summary, current works show that IL-17 plays a key role in SSc and that it is involved in the development of the disease, representing a therapeutic target of great interest.

### 3.3. Sjögren Syndrome

Sjögren’s syndrome (SS) is an autoimmune disease characterized by inflammation of the salivary, lacrimal and other exocrine glands, leading to symptoms that are a consequence of dryness, particularly in the eyes and mouth. SS patients present other clinical manifestations such as arthralgia, asthenia and pulmonary, skin, renal and neurological disorders, so it is also a systemic disease [84,85]. Besides this, SS patients have a higher risk of developing B cell lymphomas [86]. SS can develop autonomously (primary SS) or in the context of other autoimmune diseases such as SLE or RA (secondary SS) [85].

Similarly to other systemic rheumatic diseases, IL-17 levels are also elevated systemically and locally in SS patients. IL-17-producing cells are increased in the salivary glands of SS patients. Among the infiltrated IL-17-producing cells found in these glands, Th17 cells are the predominant population, although other IL-17-producing cells such as CD8^+^ T cells and double-negative T cells have also been found [87,88,89,90]. As a consequence of this higher infiltration of IL-17-producing cells, IL-17 levels are elevated in the salivary glands [87,90]. Moreover, IL-17 levels and IL-17-producing cells are also elevated in the peripheral blood and ocular surfaces of SS patients compared with healthy individuals [88,89,90,91,92,93,94,95]. Interestingly, lachrymal IL-17 levels were associated with two ocular parameters (tear film break-up time (TBUT) and Schirmer’s test), suggesting a pathological role of IL-17 in SS [96,97]. Remarkably, the IL-17R expression levels are also elevated in the glandular ducts of patients with SS [98]. These results in SS patients are also supported by similar findings in mouse models of SS, in which systemic and local levels of IL-17, as well the frequency of Th17 cells, are elevated in the salivary and lacrimal glands of these mice [99,100,101]. In addition, it has been found that cytokines that promote Th17 differentiation, such as TGF-β, IL-6 and IL-23, are also elevated in SS patients [90].

Several works have demonstrated the involvement of IL-17 in the development of SS. Lin et al. found, in a mouse IL-17 knockout model, that IL-17 plays a key role in the regulation and activation of B cells in SS [102]. In this model, IL-17, alone or in combination with B-cell-activating factor (BAFF), induced the activation and maturation of B cells, promoting this manner of isotype class switching and differentiation into plasma cells, triggering the hyperproduction of antibodies and the associated damage [103]. In addition, B cell activation and maturation in SS lead to IL-6 production, promoting, in a positive feedback loop, Th17 differentiation and IL-17 production [104].

On the other hand, IL-17 has a prominent inflammatory role that triggers the induction of cytokines, chemokines and MMPs, leading to the perpetuation of inflammation and tissue damage [104]. IL-17 promotes IL-8 and IL-6 production by human salivary gland cells [87]. Different works have shown that the expression levels of MMP-3 and MMP-9 are elevated in the plasma and salivary glands of SS patients, contributing in this way to tissue damage [105,106]. As the inhibition of IL-17 reduced the expression of MMP-9 and MMP-3 in the corneal epithelial cells of a dry-eye disease animal model [107], it is tempting to speculate that IL-17 is also involved in the elevated levels of MMPs observed in SS patients.

Altogether, these data suggest that neutralization of the IL-17 signaling could be also beneficial for the treatment of SS pathology.

### 3.4. Other Systemic Rheumatic Diseases

In the recent years, different studies have demonstrated that IL-17 has a key role in the pathogenesis of other systemic rheumatic diseases, including vasculitides and immunoglobulin G4–related disease, among others.

Primary vasculitides are a group of disorders characterized by the inflammation of the blood vessels, and whose mechanisms of action and etiopathology are mostly unknown. Although primary vasculitides are a heterogeneous group of diseases, they have been classified according to the size of the blood vessels affected [108]. Giant cell arteritis (GCA) is one of the most studied and prevalent vasculitides. Specifically, it belongs to the group of large vessel vasculitides. In GCA patients, expression levels of IL-17 are elevated in the affected tissues [109]. Likewise, high circulating levels of IL-17 and IL-17-producing cells were found in GCA patients [110]. Moreover, IL-17 levels have been postulated as a predictor of the response to treatment with glucocorticoids, as patients with higher levels of IL-17 were more sensitive to glucocorticoid treatment. In contrast, it was found that increased IFNγ levels in GCA patients resulted in resistance or relapse during glucocorticoid treatment [109,111], suggesting a different and maybe opposite role of these cytokines in the pathogenesis of GCA.

Similar findings have been observed in other vasculitides, as increased levels of IL-17 were found in primary angiitis of the central nervous system [112], Kawasaki’s disease [113] and Behcet’s disease [114], as well as in other vasculitides [115].

Despite IL-17 expression being elevated in vasculitides, the functional consequences of this dysregulation are still unclear. However, the fact that IL-17 target cells, such as fibroblasts, endothelial cells, and vascular smooth muscle cells [17], are key in the development of vasculitides and these cells express IL-17 receptors, suggest that IL-17 might be responsible for the production of pro-inflammatory mediators in these diseases [110]. However, further studies are needed to elucidate the role and mechanisms of action of IL-17.

IgG4-related disease (IgG4-RD) is characterized by an increase in IgG4^+^ cells and circulating IgG4, which is associated with the development of a fibro-inflammatory process mediated by tissue infiltrations of immune cells, including IgG4-positive plasma cells. This process leads to the formation of pseudotumor structures that produce dysfunction or lesions in the affected organs [116,117]. Despite the mechanisms of action involved in this disease remaining largely unknown [118], some studies have suggested that IL-17 might be implicated in the pathogenesis of the disease.

Grados et al. [119] found that levels of IL-17, as well as the frequency of IL-17-producing cells, were increased in patients with IgG4-RD compared to control patients. Moreover, the number of T follicular helper 17 cells (T_fh_17) was higher in patients with IgG4-RD, which could contribute to the pathology of the disease by promoting activation, differentiation and IgG4 class switching in B cells [119]. However, other studies indicate that there are no differences in the number of circulating T_fh_17 cells between patients and controls and that T_fh_17 cells are not key in the activation and differentiation of B cells [120]. Therefore, further studies are needed in order to determine whether IL-17 signaling is dysregulated in this disease. Regarding the possible effect of IL-17 on this disease, it has been postulated that IL-17 could act as a profibrotic cytokine in IgG4-RD, as well as to promote inflammatory reactions in affected tissues [121], although further studies are needed in order to validate these findings.

## 4. IL-17 Inhibitors in Systemic Rheumatic Diseases

The relevance of IL-17 in rheumatic diseases has been demonstrated with the approval of three different inhibitors of the IL-17 signaling for the treatment of PsA and AS [122].

Secukinumab (AIN457) is a fully human, anti-IL-17A monoclonal antibody indicated for the treatment of moderate to severe plaque psoriasis and active psoriatic arthritis in adult patients who have shown an inadequate response to previous treatments with disease-modifying antirheumatic drugs (DMARDs), axial spondyloarthritis (AxSpA), and active AS in adults who have not responded adequately to conventional treatment and non-radiographic axial spondyloarthritis (nr-AxSpA) [123,124].

There are currently a couple of ongoing clinical trials testing the efficacy of Secukinumab on systemic rheumatic disease (Table 1):
An active and recruiting phase III trial of Secukinumab for lupus nephritis (Clinicaltrials.gov identifier: *NCT04181762*) a 2-year, phase III randomized, double-blind, parallel-group, placebo-controlled trial to evaluate the efficacy and safety of secukinumab in combination with the standard of care therapy in patients with active lupus nephritis. The primary endpoint is the proportion of subjects achieving complete renal response (CRR) [125];An active but not recruiting phase II trial of secukinumab for giant cell arteritis (Clinicaltrials.gov identifier: *NCT03765788*) randomized, parallel-group, double-blind, placebo-controlled, multicenter, Phase II study to evaluate the efficacy and safety of secukinumab to maintain disease remission up to 28 weeks including corticosteroid tapering, in patients with newly diagnosed or relapsing giant cell arteritis (GCA) who are naïve to biological therapy [125].

Brodalumab (AMG 827) is a human, anti-IL17 receptor monoclonal antibody that binds with high affinity to human IL-17 receptor A and blocks the biological activity of the pro-inflammatory cytokines IL-17A and IL-17F, resulting in the inhibition of inflammation and clinical symptoms associated with psoriasis. The clinical efficacy and safety of brodalumab have been adequately contrasted in the authorized indication of psoriasis by means of three phase-3-randomized, double-blind, placebo-controlled (two of them with ustekinumab as an active comparator) clinical trials: the AMAGINE-1, 2 and 3 [126,127].

Regarding its efficacy in rheumatic diseases, there are several clinical trials on PsA, AS and RA and, importantly, two clinical trials on SSc (Table 1):
An active but not recruiting phase III clinical trial (Clinicaltrials.gov identifier: *NCT03957681*) of brodalumab for moderate to severe systemic sclerosis: placebo-controlled, double-blind comparative study of brodalumab with an open-label extension period in subjects with systemic sclerosis who have moderate to severe skin thickening. The primary outcome measures the change in modified Rodnan skin score (mRSS) from baseline at week 24 [125];An active but not recruiting phase I clinical trial (Clinicaltrials.gov identifier: *NCT04368403*), open-label, multiple-dose study of brodalumab in subjects with systemic sclerosis. The primary outcome measures the serum concentration of brodalumab [125].

Ixekizumab (LY2439821) is a humanized monoclonal antibody (IgG4) directed against IL-17A and IL-17A/F. It is currently approved for moderate to severe plaque psoriasis in adult candidates for systemic treatments and active PsA in adult patients with an insufficient response or intolerance to one or more DMARDs [128,129]. However, the efficacy of this antibody has not been tested in systemic rheumatic diseases (Table 1).

## 5. Concluding Remarks and Future Perspectives

In this review, we summarized the role of IL-17 in systemic rheumatic diseases and the potential use of therapeutic approach. Multiple studies have shown that levels in IL-17 and IL-17-producing cells are elevated in SLE, SSc, SS, vasculitides and IgG4-related diseases. Importantly, IL-17 plays different pathological roles in these diseases, like the promotion and perpetuation of inflammation and the damage to the affected tissues. In conclusion, targeting IL-17 signaling might be a potential option in the treatment of systemic rheumatic diseases, which is highly important due to the need for new therapeutic options for these diseases, as an effective therapy is lacking for SLE, SSc and SS patients. The most common therapies are broad-spectrum immunosuppressive drugs, which have moderate to severe side effects and therapies that only treat organ manifestations or simply relieve the clinical symptoms [49,130,131,132,133]. However, the ongoing clinical trials with different IL-17 inhibitors need to be finished before concluding whether IL-17 blocking is an effective therapeutic strategy for these diseases.

## Figures and Tables

**Table 1 ijms-21-07100-t001:** Ongoing clinical trials on systemic rheumatic diseases.

Drug	Mechanism of Action	Description	Current Trials in Systemic Rheumatic Diseases	Trial Identifier
Secukinumab	Anti-IL-17A	Human Monoclonal antibody	Phase II trial in Giant cell arteritis	*NCT03765788*
			Phase III trial in Lupus nephritis	*NCT04181762*
Brodalumab	Anti-IL-17RA	Human Monoclonal antibody	Phase III trial in Systemic sclerosis	*NCT03957681*
			Phase I trial in Systemic sclerosis	*NCT04368403*
Ixekizumab	Anti-IL-17A and IL-17A/F	Humanized Monoclonal antibody	-	-

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
