# Peer review of "Blocking IL-17: A Promising Strategy in the Treatment of Systemic Rheumatic Diseases"

_ijms, 2020, doi:10.3390/ijms21197100_

Round 1

Reviewer 1 Report

Here the authors review evidence indicating or suggesting a role for IL-17A in the pathogenesis of several systemic autoimmune disorders and the potential relevance of blocking IL-17A as therapeutic strategy in these diseases.

It seems to me that this review presents the problematic of IL-17A blockade optimistically in terms of potential for the treatment of the discussed conditions.

As stressed by the authors, IL17A is a proinflammatory cytokine. However, as many biological agents, the effects of IL17 may be directly or indirectly counterbalanced by itself or other mediators. Thus,  the net outcome of IL-17A blockade is difficult to anticipate in complex clinical situations. Until well designed clinical trials will be completed and published no conclusions can be drawn on its "potential" efficacy. I would urge the authors to add a final paragraph in which they should highlight the objective limitations for reaching firm conclusions.

Specific points

Ref 58 on line 128 support the contention that IL17A serum levels will predict response to treatment in lupus nephritis. This seems to me difficult to understand or interpret. The authors of ref 58 write that: "the optimal cutoff level of IL-17 for disease activity was 19.7 pg/ml, with 93.3% sensitivity, 92.9% specificity, 90.3 PPV and AUC 0.95 (95% CI of 0.90–1)" and as predictor of response to treatment in LN: "appropriate cutoff value of IL-17 was 27.0 pg/ml, which had 91.7% sensitivity, 72.7% specificity, 68.8 and AUC 0.82 (95% CI of 0.66–0.98)". In other words, higher levels of IL17 predict response to treatment than the IL17 levels associated to disease activity. If this were true, my interpretation could be that IL17 favoring responses to treatment protects against  severe renal disease or at least against renal disease refractory to treatment. The authors should provide a critical view of the reference conclusions.

Lines 164 to 188. Here the authors summarize evidence linking IL-17A to fibrosis in SSc, which is a difficult topic. I believe that the authors should distinguish between studies conducted in rodents form those in humans. It is very clear that IL17A has a pro-fibrotic role in several murine models of fibrosis. This is strongly supported by published data. However, in human studies the bulk of published evidence is against such a conclusion. The limitation of studies in humans is due to the fact that "in vitro" models do not capture by definition the in vivo complexities. Nonetheless, the more complex human models in which full skin is submitted to the influence of IL17 do not support a pro-fibrotic role for IL17 (see J Invest Dermatol. 2020 Jan;140(1):103-112.e8. doi:10.1016/j.jid.2019.05.026.). The authors should highlight differences in the role of IL17A in murine versus human models of fibrosis.

LInes 255-256. Here the authors indicate that the presence of IL17 may predict response to GC treatment. It would be appropriate to specify that this is contrasting with the presence of IFNgamma resulting in resistance to or relapse during GC treatment.

Line 268. Here it is said that IgG4 "lead" to to the development of a fibro-inflammatory processes. I would prefer IgG4 "are associated" with..., since there is no proof for the role of IgG4 causing the disease.

Lines 347-348. Maybe the most critical point. Here the authors write:"Phase I and II clinical trials have shown promising results in SSc and lupus nephritis" to support the contention that IL17 blockade would be a wonderful strategy in these conditions. However, I am not aware that such phase I and phase II studies have been published and, apparently, the phase III studies listed in table 1 for LN and SSc are not recruiting. Thus, a would strongly suggest a word of caution in reporting this information.

Author Response

Here the authors review evidence indicating or suggesting a role for IL-17A in the pathogenesis of several systemic autoimmune disorders and the potential relevance of blocking IL-17A as therapeutic strategy in these diseases. It seems to me that this review presents the problematic of IL-17A blockade optimistically in terms of potential for the treatment of the discussed conditions. As stressed by the authors, IL17A is a proinflammatory cytokine. However, as many biological agents, the effects of IL17 may be directly or indirectly counterbalanced by itself or other mediators. Thus, the net outcome of IL-17A blockade is difficult to anticipate in complex clinical situations. Until well designed clinical trials will be completed and published no conclusions can be drawn on its "potential" efficacy. I would urge the authors to add a final paragraph in which they should highlight the objective limitations for reaching firm conclusions.

We thank the reviewer for his/her positive statements about our work. We agree that the clinical trials need to be completed before drawing any conclusion and therefore our statements are too optimistic. Therefore we have adapted the text accordingly.

Specific points
Ref 58 on line 128 support the contention that IL17A serum levels will predict response to treatment in lupus nephritis. This seems to me difficult to understand or interpret. The authors of ref 58 write that: "the optimal cutoff level of IL-17 for disease activity was 19.7 pg/ml, with 93.3% sensitivity, 92.9% specificity, 90.3 PPV and AUC 0.95 (95% CI of 0.90–1)" and as predictor of response to treatment in LN: "appropriate cutoff value of IL-17 was 27.0 pg/ml, which had 91.7% sensitivity, 72.7% specificity, 68.8 and AUC 0.82 (95% CI of 0.66–0.98)". In other words, higher levels of IL17 predict response to treatment than the IL17 levels associated to disease activity. If this were true, my interpretation could be that IL17 favoring responses to treatment protects against  severe renal disease or at least against renal disease refractory to treatment. The authors should provide a critical view of the reference conclusions.

After a carefully reading of the manuscript we agree with the reviewer. Therefore we remove the sentence and in the new version we mention that Il-17 levels might be a biomarker of disease activity, but due to contradictory works more studies are needed.

Lines 164 to 188. Here the authors summarize evidence linking IL-17A to fibrosis in SSc, which is a difficult topic. I believe that the authors should distinguish between studies conducted in rodents form those in humans. It is very clear that IL17A has a pro-fibrotic role in several murine models of fibrosis. This is strongly supported by published data. However, in human studies the bulk of published evidence is against such a conclusion. The limitation of studies in humans is due to the fact that "in vitro" models do not capture by definition the in vivo complexities. Nonetheless, the more complex human models in which full skin is submitted to the influence of IL17 do not support a pro-fibrotic role for IL17 (see J Invest Dermatol. 2020 Jan;140(1):103-112.e8. doi:10.1016/j.jid.2019.05.026.). The authors should highlight differences in the role of IL17A in murine versus human models of fibrosis.

We apologies for these unclarities, we have now clearly distinguished the mouse and human studies and we have highlighted the differences between these studies.

LInes 255-256. Here the authors indicate that the presence of IL17 may predict response to GC treatment. It would be appropriate to specify that this is contrasting with the presence of IFNgamma resulting in resistance to or relapse during GC treatment.

Thank you for the suggestion, we have corrected accordingly.

Line 268. Here it is said that IgG4 "lead" to to the development of a fibro-inflammatory processes. I would prefer IgG4 "are associated" with..., since there is no proof for the role of IgG4 causing the disease.

Thanks again for the suggestion, we have modified the sentence.

Lines 347-348. Maybe the most critical point. Here the authors write:"Phase I and II clinical trials have shown promising results in SSc and lupus nephritis" to support the contention that IL17 blockade would be a wonderful strategy in these conditions. However, I am not aware that such phase I and phase II studies have been published and, apparently, the phase III studies listed in table 1 for LN and SSc are not recruiting. Thus, a would strongly suggest a word of caution in reporting this information.

We agree with the reviewer, we have overestimated the therapeutic potential of IL-17 and, as mentioned above, the ongoing clinical trials need to be completed before the clinical trials need to be completed before drawing any conclusion. Therefore we have modified the paragraph and we have made a more cautious and realistic statement.

Reviewer 2 Report

The manuscript is interesting and well written. However, I suggest to complete the paper discussing and adding as reference the paper by Murdaca et al concerning Th17 and chronic iflammatory immune-mediated diseases.

Author Response

The manuscript is interesting and well written. However, I suggest to complete the paper discussing and adding as reference the paper by Murdaca et al concerning Th17 and chronic iflammatory immune-mediated diseases.

We thank the reviewer for his/her positive statements about our work. We have discussed and added the suggested reference in the new version of the review.

Round 2

Reviewer 1 Report

NA